# Translational Approaches with Antioxidant Phytochemicals against Alcohol-Mediated Oxidative Stress, Gut Dysbiosis, Intestinal Barrier Dysfunction, and Fatty Liver Disease

**DOI:** 10.3390/antiox10030384

**Published:** 2021-03-04

**Authors:** Jacob W. Ballway, Byoung-Joon Song

**Affiliations:** Section of Molecular Pharmacology and Toxicology, National Institute on Alcohol Abuse and Alcoholism, 9000 Rockville Pike, Bethesda, MD 20892, USA

**Keywords:** gut microbiome, dysbiosis, leaky gut, endotoxemia, fatty liver disease, ethanol, oxidative stress, inflammation, phytochemicals, antioxidant

## Abstract

Emerging data demonstrate the important roles of altered gut microbiomes (dysbiosis) in many disease states in the peripheral tissues and the central nervous system. Gut dysbiosis with decreased ratios of Bacteroidetes/Firmicutes and other changes are reported to be caused by many disease states and various environmental factors, such as ethanol (e.g., alcohol drinking), Western-style high-fat diets, high fructose, etc. It is also caused by genetic factors, including genetic polymorphisms and epigenetic changes in different individuals. Gut dysbiosis, impaired intestinal barrier function, and elevated serum endotoxin levels can be observed in human patients and/or experimental rodent models exposed to these factors or with certain disease states. However, gut dysbiosis and leaky gut can be normalized through lifestyle alterations such as increased consumption of healthy diets with various fruits and vegetables containing many different kinds of antioxidant phytochemicals. In this review, we describe the mechanisms of gut dysbiosis, leaky gut, endotoxemia, and fatty liver disease with a specific focus on the alcohol-associated pathways. We also mention translational approaches by discussing the benefits of many antioxidant phytochemicals and/or their metabolites against alcohol-mediated oxidative stress, gut dysbiosis, intestinal barrier dysfunction, and fatty liver disease.

## 1. Introduction

Ample research conducted over the past decade has revealed the expansive and critical role of the human microbiota in numerous physiological processes and pathological consequences. Before elaborating further on these microbial communities and their abundance, the terms microbiota and microbiome should first be distinguished, considering they are sometimes used interchangeably. A microbiota specifically refers to the “assemblage of microorganisms present in a defined environment” [1], while the microbiome includes “the microorganisms (i.e., bacteria, archaea, lower and higher eukaryotes, and viruses), their genomes (i.e., genes), and the surrounding environmental conditions” [1].

Various microbiomes within the human body have been well defined, including the gastrointestinal (GI) tract, lung, skin, urinary, and oral microbiomes, and collectively amount to trillions of bacterial cells that work in concert with (or sometimes in opposition to) human cells and experimental rodent models [2,3]. Current estimates suggest that 1.3 bacterial cells are present for every 1 human cell, contrary to previous suggestions of a 10:1 bacterial to human cell ratio [4]. However, these reduced estimates should not understate the breadth of microbial influence on bodily habitats. Despite differences in microbial composition among bodily habitats, the various microbiomes often perform similar functions, such as immune regulation in the GI tract, respiratory, oral, skin, and urinary microbiomes and promoting nutrient availability in the skin and oral microbiomes [2]. Yet, while bacterial–human symbiosis can occur, so too can antagonistic interactions arise, resulting in diseases, such as periodontal disease (oral microbiome), urinary tract infection (UTI) (urinary tract microbiome), and pharyngitis and pneumonia (respiratory tract microbiome), to name several [2,3]. This review briefly addresses the nature of one of the most significant microbiomes: the gut microbiome. We also describe the mechanisms of the alcohol-mediated changes in gut microbiota, leaky gut, endotoxemia, and fatty liver disease. Based on the mechanistic insights, we finally propose the translational applications by describing the benefits of many antioxidant phytochemicals and/or their metabolites in preventing alcohol-mediated oxidative stress, gut leakiness, and fatty liver disease via changing gut microbiota.

## 2. The Gut Microbiome

### 2.1. Microbiome Present in Many Tissues

It is known that different parts of the GI tract may vary in bacterial and fungal composition and abundance, with the vast majority of gut microbes colonizing the colon [5,6,7,8]. The gut microbiota encompasses all the microbes found within the human GI tract, which stretches from the mouth, stomach, duodenum, jejunum, and ileum to the end of the large intestine (colon) at the rectum and anal canal. Among all the bacterial phylum present in the gut microbiota, the majority of bacteria fall into three phyla, Bacteroidetes, Firmicutes, or Actinobacteria, with a heavy slant toward Bacteroidetes and Firmicutes [9,10]. The major genera of these phyla found in gut microbial samples include *Bacteroides*, *Alistipes*, and *Prevotella* from the Bacteroidetes phylum, *Faecalibacterium*, and *Ruminococcus* from the Firmicutes phylum, and *Bifidobacterium* and *Collinsella* from the Actinobacteria phylum [9]. Three main enterotypes/clusters, derived from three of these specific genera, were developed to classify gut microbiotas based on their bacterial composition and their energy metabolism tendencies and capacities: Bacteroides (Enterotype 1), Prevotella (Enterotype 2) and Ruminococcus (Enterotype 3) [9,10]. However, as suggested in the study, usage of human fecal samples alone does not necessarily provide a comprehensive catalog of the gut microbial abundance and composition [9,11].

Indeed, a variety of tissues/organs lie between the mouth and anal canal, including the esophagus, stomach, and small and large intestines. Unsurprisingly, the physiological environments of the various organs/tissues of the GI tract vary considerably. Thus, although the gut microbiota is considered less diverse than other bodily microbiomes [12], microbial diversity does exist among the subsections of the GI tract. Indeed, sequencing of bacterial 16S ribosomal RNA from various regions of the GI tracts in healthy fasting adults revealed distinct microbial communities in the saliva, upper GI tract, lower GI tract, and feces [13]. For example, though *Bacteroides uniformis* levels in the salivary and upper GI regions were minimal, a significant increase was detected in the lower GI tract, which further increased in feces [13]. Additionally, diversity and heterogeneity of microbial communities are noted to decrease in regions farther down the digestive tract, owing to the increase in selective pressures and environmental changes present between the stomach and intestinal regions [13]. For instance, *Prevotella melaninogenica* abundance was high in the salivary region; yet, an insignificant decrease was noted toward the end of the upper GI tract, followed by a significant reduction in the lower GI tract, where it was nearly absent [13].

Taking a closer look at the lower GI tract, a marked environmental contrast exists between the acidic, oxygen- and antimicrobial-rich small intestine, and the more basic, oxygen- and antimicrobial-depleted large intestine (colon) [11]. Facultative anaerobes from the *Lactobacillaceae* (*Firmicutes* phylum) and *Enterobacteriaceae* (*Proteobacteria* phylum) families colonize the harsh small intestinal conditions, while anaerobes from families, such as *Bacteroidaceae (Bacteriodetes* phylum) and *Lachnospiraceae (Firmicutes* phylum), inhabit the large intestine in great numbers, owing to the tolerant conditions of the colon [11,14]. Additional factors determining microbial composition of the small intestine and colon include differences in number, amount, and composition of mucus layers and nutrient availability (reviewed extensively in [11]).

Naturally, these differences in the bacterial composition suggest distinct functions for microbes at these particular regions. Indeed, in the intestines, the gut microbiota serves an important role in the metabolism of various endogenous and exogenous compounds, the catabolism of complex carbohydrates, amino acids, and fatty acids into smaller molecules, the synthesis of vitamins or short chain fatty acids (SCFAs), and the degradation of bile acids for the benefit of the host [2,11,12,15]. By the same token, it is also possible that a few microorganisms can metabolize some substrates and produce potentially harmful compounds, such as ethanol [16,17,18] and trimethylamine (TMA), or disturb the balance of certain secondary bile acids [19,20,21]. However, many metabolites of this microbial-mediated metabolism, especially SCFAs, are essential for host energy and signaling pathways as well as epigenetic regulation [2,11,12,15,19]. Additionally, the gut microbiota plays a role in immune regulation and host defense [15]. Specifically, the gut microbiota and its metabolites were shown to regulate T-lymphocytes and macrophages, interact with the enteric nervous system (e.g., by producing and/or increasing production of serotonin and melatonin), and mitigates pathogenic bacterial colonization [15,22,23,24,25]. Importantly, these functions may be altered or even compromised when changes to the composition of the gut microbiota take place. Indeed, the gut microbiota composition can vary not only between individuals and throughout the different stages of an individual’s life, especially during infancy (thoroughly reviewed in [10]), but also in response to numerous environmental factors or other pathophysiological states.

### 2.2. Gut Microbiota Altered by Aging, Disease States, and Environmental Factors

Changes in the gut microbiota due to normal aging or aging-related pathological states have been well described, and a significant portion of age-related changes occur during early human life [10,26]. Beginning with the transfer of maternal microbes during fetal development, numerous factors will subsequently alter and reshape the human gut microbiota, including the type of birth delivery, the method of milk administration, the weaning of the infant from milk onto solid food, and the interactions with family and the geographical environment during early development [26] (see [10] for a comprehensive list of microbial changes during development). Eventually, the fluctuations experienced by early gut microbiota as a result of these factors will give way to the adult gut microbiota, which face an array of environmental pressures [27]. Even so, the adult microbiota continues to fluctuate with continued aging, especially considering the results of one study, where notable differences in microbial composition, such as a significant increase in *Bacteroidetes* phylum members in elderly individuals compared to young adults, were observed [28]. Additionally, not only does the composition of the gut microbiota change in aging, but also the effectiveness of the microbiota in performing important functions. For instance, although the diversity of microbes actively synthesizing proteins was increased in elderly individuals, the levels of proteins involved in tryptophan and indole metabolism (TnaA and TrpB) were decreased in elderly people compared to infants. In fact, a predicted ~90% drop-off in tryptophan and indole production was observed for individuals 34 years old or above, which may alter immunological and neurological functioning [29].

In addition to normal aging and related physiological changes, pathological states can also play a role in altering the gut microbiota composition. Traumatic brain injury (TBI) represents one such example, and a recent study examining the fecal microbiome of individuals who have suffered chronic TBI (examined years after the acute incident) found significant decreases in *Prevotella* and *Bacteroides* species in addition to increases in the *Ruminococcaceae* family of bacteria, when compared to healthy individuals [30]. Additionally, the researchers suspect that the increases in *Ruminococcaceae* members, coupled with decreases in *Prevotella* species, specifically *Prevotella copri*, may explain the changes in amino acid metabolism and inflammation experienced by these individuals [30]. However, another study examining repetitive, mild TBI found no major alterations in the gut microbiota composition. Interestingly, an increase in the percent relative abundance of members from the *Desulfovibrionaceae* family was observed, which, as mentioned by the researchers, has been connected to cognitive impairments in previous studies, though no direct effect was evaluated or confirmed in this study [30,31].

Moreover, gut dysbiosis can also be affected by environmental factors such as various diets, including Western-style high-fat diets (HFDs) [32] containing fructose [33,34] or predominantly fructose alone [35]. Western-style HFDs containing n-6 polyunsaturated fatty acids can drive the incidence of obesity [36], which may progress to worse complications (e.g., insulin resistance, hypertension, etc.) characteristic of metabolic syndrome [37,38] or even type 2 diabetes. Western-style HFDs impact the resident gut microbiota, contributing to gut leakiness, lipopolysaccharide (LPS) translocation, and inflammatory damage in host intestinal tissue [39,40]. Specifically, a Western-style HFD appears to reduce the presence of carbohydrate-metabolizing proteins, in favor of proteins implicated in amino acid metabolism, with decreased amounts of the Ruminococcaceae family members, since they are involved in carbohydrate degradation [41]. Importantly, the increased abundance of members of the Proteobacteria phylum following HFD exposure to mice [42] is also physiologically important, considering that members of this phylum produce endotoxin LPS, which can enter host circulation due to HFD-mediated damage to host intestinal barrier and stimulate inflammatory damage in the GI tract and other tissues [39]. Interestingly, both Western-style high-fat and fructose diets can increase the abundance of members of the Proteobacteria phylum with a corresponding decrease in Bacteroidetes members, which, as previously stated, likely contributes to the increased leaky gut and LPS translocation [39,43]. Furthermore, chronic and binge alcohol (ethanol) consumption or exposure also affects the amount and composition of gut microbiomes [44,45], resulting in increased oxidative stress, intestinal permeability, endotoxemia, and damage in many tissues. In addition, Chen et al. [46] recently showed that high mobility group box-1 (HMGB1) contained in extracellular vesicles (exosomes) derived from gut dysbiosis can also promote non-alcoholic fatty liver disease (NAFLD) in adapter protein ASC-null mice after exposure to an HFD.

All these conditions clearly indicate the important role of changes in the amounts and composition of the gut microbiota, resulting in impairment or damage in various tissues or organs through the gut–liver–brain axis. In this review, we specifically focus on the role of alcohol-mediated oxidative stress in promoting intestinal barrier dysfunction, and fatty liver disease. In addition, we explain the mechanisms of alcohol-mediated gut dysbiosis, resulting in increased leaky gut and fatty liver (steatosis) and/or inflammation (steatohepatitis) via increased oxidative and nitrosative/nitrative stress. Finally, we briefly describe the beneficial effects of various antioxidant phytochemicals and their mechanisms of action against gut dysbiosis, intestinal barrier dysfunction, and fatty liver disease.

## 3. Oxidative Alcohol Metabolism and Progression to Alcoholic Liver Disease

Following consumption, most alcohol (ethanol) molecules can be oxidatively metabolized to acetaldehyde and then irreversibly converted to acetate in numerous organs/tissues, including the liver, stomach, and possibly brain at very low levels. However, the primary site of oxidative alcohol metabolism occurs in the hepatocytes of the liver [47,48], although the stomach is also known to be involved in alcohol metabolism [49]. In addition, alcohol can be metabolized by the non-oxidative metabolic pathway such as fatty acid ethyl esters by cholesterol esterase, etc. [50,51]. Furthermore, unmetabolized ethanol can be excreted from the body through the breath, skin, sweat, and urine [52].

Owing to its low Km and high expression in the liver, cytoplasmic alcohol dehydrogenase (ADH) oxidatively metabolizes the majority of alcohol entering the liver into acetaldehyde, a highly reactive intermediate, while simultaneously reducing the cofactor NAD^+^ to NADH [47,48]. The reactive (and potentially harmful) intermediate acetaldehyde is then oxidized in the mitochondria by the low Km aldehyde dehydrogenase 2 (ALDH2) with the reduction of a cofactor NAD^+^ to NADH, to generate acetate, which is converted into acetyl-CoA for metabolic use or subsequently exported to other organs, including the brain [47,48]. The ADH- and ALDH-mediated oxidative ethanol metabolism pathway provides a reliable and efficient means of metabolizing the majority of consumed alcohol. Aside from the physiological consequences of alcohol overconsumption, social problems arise when alcohol consumption increases in frequency and amount. In fact, more than ~75% of alcohol-associated with sociomedical consequences are ascribed to the consumption of large amounts of alcohol in short periods of time [53,54].

Naturally, increased amount and frequency of alcohol consumption begins to deplete the cellular levels of the NAD^+^ cofactor by the ADH- and ALDH2-dependent reactions. A decrease in NAD^+^ levels, coupled with an increase of NADH levels, will cause redox changes and alter numerous cellular functions, resulting in elevated fat synthesis, decreased fat oxidation, and increased cell death processes, among others, with characteristic hallmarks of early alcoholic liver disease (ALD), specifically, steatosis and liver inflammation [47,55]. Furthermore, two additional enzymes are involved in oxidizing ethanol in the liver: peroxisome-resident catalase and endoplasmic reticulum (ER)- and mitochondria-localized ethanol-inducible cytochrome P450-2E1 (CYP2E1), which represents a major component of the microsomal ethanol oxidizing system (MEOS) [48,56]. However, the role of catalase in hepatic ethanol metabolism is minor in comparison possibly due to the limited availability of hydrogen peroxide [56]. Thus, the CYP2E1 enzyme, constitutively expressed under normal physiological states and then induced by ethanol, at least via protein stabilization [57,58], becomes functionally important in ethanol metabolism, as well as in alcohol-mediated oxidative liver damage, particularly because CYP2E1-mediated alcohol metabolism results in the production of both highly reactive acetaldehyde and reactive oxygen species (ROS), such as the superoxide anion (O_2_^−^) and hydrogen peroxide (H_2_O_2_) [47,59,60]. In fact, *Cyp2e1*-null mice are resistant to alcohol-mediated liver injury [61], while transgenic mice with overexpressed CYP2E1 [62] and *Cyp2e1* knock-in mice were more sensitive to liver injury by alcohol [63] or non-alcoholic substances, including a diet with 20% fat-derived calories [64]. Overwhelming increases in ROS levels strain the cellular antioxidant defense mechanisms, resulting in elevated oxidative stress, various post-translational protein modifications (PTMs), and apoptotic cellular damage through increased lipid peroxidation, ER stress, mitochondrial dysfunction, and DNA damage with genomic instability [47,65,66,67,68,69]. If alcohol consumption persists, hepatic damage will continue to increase and prime the liver for progression into more severe stages of ALD such as liver fibrosis, cirrhosis, and hepatocarcinoma [70,71].

ALD pathogenesis has been well reviewed elsewhere [72,73,74,75]; however, here we will briefly address the main phases of ALD and the pathological hallmarks of each stage that will be relevant when discussing the contributing role of the gut microbiota in ALD pathogenesis. The first major stage of ALD is the development of fatty liver, also termed steatosis, which involves the accumulation of lipid droplets within hepatocytes. The mechanisms underlying the development of alcohol-induced fat accumulation are numerous. Important mechanisms include a redox change with the decreased NAD^+^/NADH ratio during the oxidative ethanol metabolism and increased fat synthesis in the cytoplasm, through activated transcription factors, such as sterol regulatory element binding protein (SREBP-1c). In addition, ethanol intake can cause fat accumulation via decreased fat degradation, resulting from the suppressed mitochondrial enzymes for the fat oxidation pathway and acetaldehyde-mediated decreased transcription of peroxisome proliferator activator receptor α (PPARα) needed for fatty acid export and degradation, and increased import/transport of free fatty acids from adipose tissues after lipolysis. Persistent fat accumulation and oxidative stress can severely damage hepatocytes, resulting in apoptosis, which promotes the activation of liver-resident macrophages (Kupffer cells) and attracts infiltrating neutrophils, leading to inflammation and steatohepatitis. During steatohepatitis, Kupffer cells and other liver cells respond to both damage-associated molecular pattern (DAMP) molecules from apoptotic hepatocytes, in addition to other molecules, such as gut-derived pathogen-associated molecular pattern (PAMP) molecules such as LPS, leading to the further secretion of proinflammatory cytokines and the persistence of oxidative stress, thus exacerbating liver damage [47,76]. Eventually, hepatic stellate cells can be activated by transforming growth factor-β (TGF-β) secreted by Kupffer cells attempting to resolve inflammation, which may propel the liver toward fibrosis [47]. Liver fibrosis arises as structural proteins, such as collagen and α-smooth muscle actin (α-SMA) derived from transformed hepatic stellate cells, assemble into the extracellular matrix to form a network of rigid, fibrotic scar tissue to surround damaged portions of the liver [77]. Acetaldehyde, produced during the oxidative ethanol metabolism and potentially elevated due to inactivation of ALDH2 under oxidative stress [65,78,79,80], is recognized to modulate important aspects of fibrosis, such as inhibiting PPARγ or increasing the transcriptional activity of C/EBPβ to stimulate collagen α1(I) expression [67]. Persistent fibrosis leads to liver failure during cirrhosis, as unresolved fibrotic tissue continues to damage liver architecture and hinders liver recovery and function [47,74]. Development of hepatocellular carcinoma can arise not only from the formation of various adducts between DNA or proteins and acetaldehyde, malondialdehyde (MDA), or 4-hydroxynonenal (4-HNE) (likely resulting at least partially from CYP2E1-mediated oxidative stress) [81], but also from the release of PAMP and DAMP molecules, that can activate immune cells and indirectly contribute to the abundance and increased activity of tumor-initiating stem-cell-like cells (TICs) [74].

With this basic overview of ALD in mind, we can now analyze the direct effect of alcohol on the gut function and the gut microbiota. Following this analysis, we will systematically address the effects of alcohol-induced changes to the gut microbiota during ALD. The contribution of specific bacterial groups and/or species to liver and gut damage following alcohol exposure will be discussed. Additionally, this review will highlight some of the numerous therapeutic options that may mitigate alcohol-induced oxidative stress, gut dysbiosis, leaky gut, and fatty liver by various dietary supplements, such as antioxidant phytochemicals, probiotics, small molecule metabolites, and traditional/ancient medications.

## 4. The Mechanisms of Alcohol-Mediated Gut Dysbiosis, Intestinal Barrier Dysfunction, and Consequences

Over the past decade, several well-published articles have reviewed the role of alcohol-induced gut dysbiosis on ALD pathogenesis [20,82,83,84,85]. In fact, it has been reported that people with ALD, including liver cirrhosis, have elevated levels of serum endotoxin compared to control subjects [44], indicating increased intestinal permeability or gut leakiness (leaky gut). This seminal observation with AUD people was replicated by many other laboratories [86,87,88]. Furthermore, the elevated gut dysbiosis and leaky gut following long-term and/or binge ethanol exposure was also observed in experimental models with mice [89] and rats [86,90], indicating a common phenomenon conserved among different species. The following section will highlight alcohol-induced gut dysbiosis, the specific changes in their amounts and composition during ALD, and potential translational approaches designed to remedy these alterations.

### 4.1. The Effect of Alcohol on the Amounts and Composition of Gut Microbiota

Many microbes are affected by the presence of alcohol in the various parts of the GI tract, and the changes in the abundances and composition of the gut microbiota have been extensively studied. Conveniently, a recent study [91] has compiled data from many publications describing changes in the gut microbiota in humans, such as individuals with AUD [92], those who have a history of chronic overconsumption of alcohol [93], and those in different stages of ALD [94]. Examining the data from a subset of these publications (in addition to several very recent publications) reveals important trends to consider for ALD and gut dysbiosis prevention.

Alcohol intake is known to increase the degree of small intestine bacterial overgrowth (SIBO), as originally reported [44,45,95]. In addition, several important phyla, including the major *Proteobacteria*, *Bacteroidetes*, *Firmicutes*, and *Actinobacteria*, are all impacted by the presence of alcohol in the GI tract. Numerous studies examining human colonic biopsies [96] and human [93] and mouse [97] feces indicate a higher abundance of members of the *Proteobacteria* phylum in response to alcohol [91]. As suggested elsewhere, this change can conceivably result from the ability of microbes in this phylum to persist in the high ROS environment generated by increased inducible nitric oxide synthase (iNOS) following alcohol exposure [98], since they are predominantly facultative anaerobes, which can withstand these conditions [92]. Specifically, at the family level, *Enterobacteriaceae* abundance was increased [91,92,96], in addition to elevated levels of certain genera from this family, such as *Escherichia* [91,92,96]. Owing to their Gram-negative status and endotoxin (e.g., LPS) producing capabilities, *Proteobacteria*, such as those from the genera *Escherichia*, are unsurprisingly seen as potential instigators of gut barrier dysfunction during alcohol consumption [93,97]. In particular, the ability of species in the *Escherichia* genus to produce harmful PAMP molecules, such as LPS [93], and to metabolize alcohol (in some strains) [99], producing the toxic metabolite acetaldehyde, supports the hypothesized harmful role of *Proteobacteria* in alcohol-induced gut dysbiosis. Additionally, although the *Proteobacteria* genera *Sutterella* displayed a decreased relative abundance in the colonic biopsies of heavy drinkers [96], another study examining the feces of individuals with a history of chronic alcohol overconsumption found a higher relative abundance of this genera and views the *Sutterella* increase in light of previous studies reporting its role in promoting inflammation [93]. This same study provides evidence for an inverse relationship between *Proteobacteria* and the presence of the anti-inflammatory SCFA butyric acid, which was decreased in the feces of individuals with a history of chronic AUD, although as noted by the authors, confirming this correlation is hindered by the nature of the study [93].

Alcohol exposure has been shown to decrease the abundance of members of the phylum *Bacteroidetes* in the feces of mice exposed to chronic alcohol [97] and from colonic biopsies [87] and feces [92] of AUD individuals. However, several other reports using chronic alcohol mouse models have described an increase in *Bacteroidetes* presence following the sequencing of colonic and cecal contents [20,100,101]. More conflicting results emerge when examining the genus *Bacteroides* in this phylum. Although one recent study did not detect any difference in the abundance of *Bacteroides* members in colonic biopsies from heavy drinkers [96], other studies examining stool from AUD patients found an increase in *Bacteroides* members [102], while the sequencing of feces from alcoholic individuals [92] found decreased abundance of this genus. As mentioned by the authors, unlike facultative anaerobic *Proteobacteria, Bacteroides* members are obligate anaerobes and may, therefore, struggle to survive in the presence of significant ROS during prolonged alcohol exposure, which may explain the decrease in this study [92]. Although *Bacteroides* are not active ethanol metabolizers [92], they are known to play a role in the metabolism of bile acid, which could interfere with farnesoid X receptor (FXR) signaling (due to its regulation by bile acids) [20]; however, elevated Bacteroides levels could lead to increased production of the gamma-aminobutyric acid (GABA) neurotransmitter [102], suggesting a potential interplay between members of the *Bacteroides* genus and the brain during ALD. Interestingly, one study showed that members of another genus, *Prevotella*, have the capacity to metabolize ethanol and generate acetaldehyde in vitro, suggesting a possible role for members of this genus in contributing to acetaldehyde production in vivo [92]. However, like the *Bacteroides* data, conflicting reports have emerged regarding the relative abundance of *Prevotella* during alcohol exposure, where *Prevotella* members displayed increased abundance in the stool of AUD patients [102]; yet, numerous studies have described a decrease in the relative abundance of members from this genus [91].

In a mouse model, the presence of members of the *Actinobacteria* phylum was noted to increase in the feces of alcohol-exposed mice, and this elevation, in conjunction with the amplified presence of *Proteobacteria,* has been suggested to play a role in the intestinal manifestations of ALD [97]. Indeed, several interesting genera from this Gram-positive phylum are altered in the gut following alcohol exposure, including *Corynebacterium, Bifidobacterium*, and *Collinsella* [91,97]. *Corynebacterium* was found to be increased in the feces of mice chronically exposed to alcohol and, although the relevance of this elevation in ALD pathogenesis remains unknown, the authors posit that this amplification could be noteworthy, since other studies have found *Corynebacterium* infection in ALD [97,103,104]. The other genera, *Bifidobacterium* and *Collinsella*, display conflicting alterations among studies. While *Bifidobacterium* was decreased in human feces of individuals who habitually drink alcohol [92], this genus was found in increased abundance in the feces of active alcoholic patients with cirrhosis and severe alcoholic hepatitis, compared to individuals with cirrhosis and lacking severe alcoholic hepatitis [94]. Similarly, while *Collinsella* was found in increased abundance in the stool of AUD individuals [102], analysis of the relative abundance of *Collinsella* showed no significant change in the abundance of these genera in the feces of individuals who habitually consume alcohol [92]. Interestingly, both *Bifidobacterium* and *Collinsella* were characterized as potential acetaldehyde accumulators in in vitro aerobic conditions [105], and, in particular, the absence of *Bifidobacterium* was hypothesized to contribute to the observed decrement in alcohol metabolism in the feces of alcoholic individuals [92]. Some have postulated that the ability for *Collinsella* to metabolize ethanol may permit their observed overgrowth in the stool of AUD patients [102], or could potentially allow them to persist longer than other microbes.

Although the phylum *Firmicutes* was demonstrated to decrease in abundance in mouse fecal samples following chronic alcohol exposure [97], this diverse phylum has several genera that were both increased, including *Streptococcus, Coprobacillus, Holdemania,* and *Clostridium*, and decreased, including *Ruminococcus, Faecalibacterium, Subdoligranulum, Roseburia*, and *Lactobacillus*, among others, following alcohol exposure [91,92,93,96,101,102]. *Roseburia* and *Lactobacillus*, two therapeutically relevant bacterial genera, showed decreased abundance in colonic biopsies of heavy drinkers [96] and colonic contents of rats chronically exposed to alcohol [101], respectively. Similarly, *Faecalibacterium* and anti-inflammatory members of the genus, such as *F. prausnitzii*, are decreased in the feces of heavy drinkers [96] and individuals who have a history of AUD [93]. Expectedly, these decreased levels of *F. prausnitzii* during alcohol exposure likely impact the levels of beneficial SCFAs in the intestines [91], such as butyrate, and, unsurprisingly, a positive correlation was observed between *Faecalibacterium* and butyric acid levels in the feces of AUD individuals [93]. Overall, the Firmicutes phylum is quite diverse. While obligate anaerobes from the Ruminococcus genus are observed to decrease [92], likely the result of the increased oxidative stress in the gut following alcohol consumption, as previously reported [98], facultative anaerobes from the Streptococcus genus have been shown to elevate in both the stool of patients with AUD [102] and in the feces of alcoholic individuals [92]. Indeed, as others have postulated, the observed decrease in obligate anaerobes may give members of the Streptococcus genus (and other facultative anaerobes) an opportunity to proliferate, which may be of concern, considering that infections from Streptococcus members have been noted during cirrhosis [91].

Lastly, from the phylum *Verrucomicrobia*, the *Akkermansia* genus was noted to decrease in both the stool of people with AUD [102] and in colonic biopsies from heavy alcohol consumers [96]. The *Akkermansia* genus and, specifically, *Akkermansia muciniphila*, have numerous beneficial roles in the intestines, including protecting the gut barrier and aiding in the production of epithelial cell-protective mucus [91,96,102], and supplementation was shown to prevent manifestations of ALD in binge and chronic mouse models of alcohol exposure [106]. Importantly, in patients with AUD, low *Akkermansia* levels negatively correlated with the inflammatory marker MCP-1, indicating a potentially significant role for these microbes in inflammation regulation during alcohol consumption [102].

Importantly, when analyzing the alcohol-induced changes in the composition of the gut microbiota, one needs to take special consideration for the specific tissue/sample being examined. For instance, examining microbial changes in the feces of mice following alcohol exposure [97] or humans who have consumed alcohol at some time [92,93] does not necessarily provide a comprehensive assessment of the gut-wide microbial status. As suggested by others, analysis of microbial changes in samples other than feces is needed to pinpoint changes in the diverse environments of the gut [91]. For example, one study examining the microbial changes in jejunal and colonic contents of rats chronically exposed to alcohol found significant changes in colonic microbiota, but hardly any impact on composition in the jejunal microbiota [101]. Thus, future research may target specific areas of the intestines (duodenal, jejunal, ileal, colonic, rectal, etc.), which should provide a more complete profile of the gut microbial changes following alcohol exposure.

### 4.2. Mechanisms of Alcohol-Induced Damage to the Intestines, Resulting in Leaky Gut and Endotoxemia

Before ethanol molecules reach bodily destinations, such as the liver and brain, they must first pass through the tissues of the GI tract. Alcohol can be absorbed in the mouth and esophagus (albeit in limited quantities), and absorption rates will begin to increase further down the GI tract, especially at the stomach, duodenum, and jejunum and, to a lesser extent, at the ileum [107,108]. Alcohol absorption at the proximal regions of the small intestine (duodenum and jejunum) [109] will result in the passage of these molecules into the capillaries and blood vessels, including the portal vein, where they will be delivered to the liver [110]. Although alcohol passes from the intestinal lumen into the bloodstream through simple diffusion [107,111], ethanol molecules first must pass through the layers of the intestinal barrier and can trigger changes at these various regions.

Considering the structure of the small and large intestinal barrier, both consist of a monolayer of intestinal epithelial cells with a layer of mucus on the luminal side and the immune cell-rich lamina propria on the non-luminal side. Aside from the chief intestinal epithelial cell, the enterocyte, the monolayer can contain numerous other cell types. These include intestinal epithelial stem cells (IESCs), which can differentiate to replace cells in the monolayer, Paneth cells, which produce antimicrobial peptides and regulate IESCs, and goblet cells, which produce and secrete mucin glycoproteins for incorporation into the luminal mucus layer [112,113]. Several differences exist between the small and large intestinal barriers. Specifically, unlike the small intestine, the colon contains two mucus layers, a loose and a thick layer [11], a proportionally greater number of goblet cells, and an absence of Paneth cells [112].

In the intestinal lumen, ethanol will encounter the mucus layer, which contains glycosylated mucin proteins and lipids, among other components, and serves as both a protective barrier and a regulator of the microbial environment, whereby species, such as *Akkermansia muciniphila*, will use barrier-generated mucus resources [11,112]. Interestingly, mRNA levels of mucin proteins 1-4 do not increase in the small or large intestines in response to binge alcohol [114]. However, examining the glycosylation status of mucin glycoproteins, studies using chronic models of alcohol exposure found altered patterns of mucin glycosylation in the intestinal mucosa, especially increased galactosylation [115], which may influence bacterial binding to the mucus layer or may prevent mucus adherence to the intestinal barrier, as demonstrated in a gastric mucosa study [116]. Importantly, one study has suggested that ethanol may also decrease the hydrophobicity of the mucus layer, specifically, through the ethanol-mediated dissolution of mucus-layer free fatty acids, which function as an absorptive barrier, thus likely promoting ethanol-induced gut permeability dysfunction [117].

Eventually, from the mucus layer, ethanol will pass into the monolayer through simple diffusion, where it will either continue to diffuse into the circulation for delivery to various bodily sites or be metabolized in the barrier [108]. Indeed, the presence of ADH and ALDH isozymes and their activities have been detected in the small and large intestines, although their contribution to alcohol metabolism is not as significant as the liver [118]. Although differences in ADH and ALDH expression vary across the intestinal landscape, importantly, ADH expression and activity appears greater in both the small and large intestines compared to those of ALDH, suggesting a greater buildup of reactive acetaldehyde over acetate in the monolayer following alcohol metabolism [118]. In addition, the presence of CYP2E1 and its elevated levels in the GI tract after alcohol exposure [119,120,121] are likely to produce elevated levels of ROS and acetaldehyde, especially due to low levels of ALDH2 expression in the gut [122,123]. As reviewed thoroughly elsewhere [108], the metabolism of ingested alcohol can also occur through non-oxidative pathways. Ingested alcohol may also interact with and be metabolized by gut microbes. Indeed, some species are capable of metabolizing ingested alcohol, such as certain strains of *Escherichia coli* [99], while other gut microbes display limited alcohol metabolism ability, such as some members of the lactobacillus genus [124]. Additionally, a recent study examining the feces of Japanese alcoholic individuals categorized several bacterial groups, such as the *Ruminococcus* and *Collinsella* genera, as potential acetaldehyde accumulators for their ability to metabolize ethanol in aerobic in vitro conditions [105,125].

With these factors in mind, along with the previous discussion of the alcohol-mediated alterations to the gut microbial composition, we can now examine the mechanisms and impact of gut dysbiosis and alcohol exposure on the gut barrier, specifically, on the increased intestinal permeability observed following chronic and/or binge alcohol exposure. Normally, the intestinal monolayer forms a tightly linked barrier with various proteins forming the intestinal tight junction (TJ), adherent junction (AJ), and desmosomes that keep microbes in the intestinal lumen out of the bloodstream [126], while also permitting the passage of luminal nutrients into the bloodstream, thus ensuring a useful, non-toxic blood supply for recipient organs, such as the liver via the portal vein [127]. However, alcohol exposure can alter the permeability of this intestinal barrier, resulting in an influx of harmful luminal molecules, such as bacterial endotoxin LPS and peptidoglycan, which can induce inflammation and oxidative damage both in the gut and in other organs [127]. Based on this information, we will concisely address the role of ethanol, acetaldehyde, ROS, and other factors in triggering this intestinal permeability dysfunction and will subsequently address the impact of this leaky barrier on the various stages of ALD pathogenesis, as illustrated in Figure 1. This review will place a particular emphasis on the role of oxidative and nitrosative/nitrative stress in gut dysfunction and its impact on ALD pathogenesis, in addition to the mechanistic role of enterocyte apoptosis and PTMs of paracellular junctional complex proteins in initiating gut barrier dysfunction (leaky gut), leading to inflammatory liver injury (Figure 1).

Early studies determined that exposure of ethanol to the human colonic Caco-2 cell line triggered apoptosis in these epithelial cells [128]. Although a mouse model of chronic ethanol exposure alone displayed no significant changes in apoptotic protein markers or intestinal permeability in the jejunum [129], binge alcohol exposure elevated endotoxin levels in rodent models, pointing to the development of a leaky gut in these animals [130,131]. The effects of binge alcohol exposure on time- and dose-dependent gut permeability change, elevated endotoxemia, and fatty liver injury were confirmed by other laboratories [121,132,133]. Mouse and rat models of binge alcohol exposure showed increased levels of apoptotic protein markers, such as BAX and cleaved caspase-3, and histological evidence for the apoptosis of intestinal epithelial cells, consequently indicating increased gut permeability [121]. In rodent models, treatment with antioxidants or a specific inhibitor of CYP2E1 significantly prevented binge alcohol-mediated leaky gut and fatty liver disease, while *Cyp2e1*-knockout mice were also quite resistant to these changes [120,121,132]. These results clearly indicate the important roles of CYP2E1 and consequent oxidative stress in promoting intestinal barrier dysfunction and ALD. Furthermore, a recent study using a mouse model of chronic plus binge alcohol exposure found increased apoptotic markers in the proximal small intestine, likely mediated by ER stress, and this corresponded to the observed increase in bacterial product translocation at this region [134]. Importantly, however, the effect of ethanol on the intestinal barrier is not limited to the induction of apoptosis in the epithelial cells within the gut monolayer.

Several important proteins that normally connect cells of the monolayer and enhance the impermeability of the intestinal barrier are affected by ethanol. To limit the travel of molecules between adjacent monolayer cells (the paracellular pathway), TJ complexes composed of proteins, such as claudins, occludin, and zonula occludens-1 (ZO-1), along with AJ complexes, composed of proteins, such as E-cadherin and β-catenin [135,136], form between monolayer cells to prevent the mass influx of particles into the bloodstream. Exposure of ethanol to Caco-2 cells resulted in a time-dependent increase in epithelial cell permeability, which correlated with the time- and dose-dependent decrease in ZO-1 and increase in claudin-1 protein levels [137]. Additionally, ZO-1 and claudin-1 were noted to be irregularly distributed upon localization. A similar observation was noted in Caco-2 cells treated with ethanol (10, 20, or 40 mM) for 3 h, whereby ZO-1 and occludin showed decreased localization to the membranes for intercellular interactions, which correlated with increased barrier permeability, despite no observed change in ZO-1 (and other tight junction proteins) mRNA levels [138]. Thus, the presence and proper localization of junctional complex proteins involved in paracellular transport are crucial for maintaining intestinal integrity, especially considering that occludin knockout mice exposed to ethanol displayed both increased permeability at the colon and increased triglyceride accumulation in the liver compared to wild-type (WT) mice exposed to ethanol [139]. In addition, ethanol and consequently CYP2E1-mediated oxidative and nitrosative/nitrative stress appear to induce changes to the PTM landscape of intestinal proteins on a global level, with regard to increased acetylation, nitration, and ubiquitination [120] as well as phosphorylation [140,141,142,143].

### 4.3. Mechanisms of Gut Leakiness via Oxidative Stress and PTMs of Paracellular Proteins

Interestingly, ethanol appears to induce changes to the PTM landscape of intestinal proteins on a global level, with regard to acetylation, oxidation, nitration, and ubiquitination [120,121]. One recent study found that the removal of intestinal NAD^+^-dependent class III deacetylase SIRT1 blunts alcohol-induced liver damage in an acute on chronic alcohol mouse model, likely through prevention of ferroptosis, which may indicate a potential role for global acetylation in intestinal barrier function in a model of ALD pathogenesis [144]. However, in particular, various PTMs of proteins involved in the blockage of paracellular transport, such as those comprising TJ, AJ, and cytoskeletal proteins, have received considerable interest for their possible role in stimulating gut leakiness following alcohol exposure. Indeed, many paracellular transport proteins, especially TJ proteins, are known to be post-translationally modified, which play a role in their functional capabilities [145,146].

In the context of alcohol exposure, acetylation of α-tubulin appears to interfere with ZO-1 recruitment to the membrane, which likely contributes to increased permeability observed in the in vitro Caco-2 cell model following exposure to ethanol or acetaldehyde [138]. Additionally, increased nitration and ubiquitin-conjugation of α-tubulin in the intestines of mice exposed to binge alcohol correlated with decreased protein levels of α-tubulin and endotoxemia in these mice [121]. Modification of specific TJ and AJ proteins has also been well documented. In this same study, increased nitration and ubiquitin-conjugation of β-catenin (AJ), plakoglobin (adherens junction/desmosomes), claudin-1 (TJ), and claudin-4 (TJ) was observed in the intestines of binge alcohol-exposed mice, which, once again, correlated with endotoxemia and, thus, gut leakiness and fatty liver disease [121]. The significantly decreased gut TJ and AJ proteins in binge alcohol-exposed rats compared to controls were further confirmed by quantitative mass-spectral analysis. Additionally, the role of paracellular protein phosphorylation in both TJ and AJ formation has been well characterized and is altered by acetaldehyde [140,147]. Specifically, the presence of acetaldehyde leads to the persistent phosphorylation of the TJ protein occludin and AJ proteins E-cadherin and β-catenin, which are hypothesized to prevent the binding of these proteins to actin filaments, evidenced by the decreased amounts of these proteins in actin-rich Triton-insoluble fraction, which, as indicated by the authors, is an accurate predicator of TJ and AJ complex integrity [148]. Further studies have pinpointed the specific enzymes involved in regulating paracellular protein phosphorylation status following alcohol exposure. The protein phosphatase 2A (PP2A)-mediated dephosphorylation of threonine residues on occludin [149] and the altered activity of protein tyrosine phosphatase 1B-mediated dephosphorylation of AJ proteins [147] are two such examples of alcohol-induced changes to the paracellular protein landscape. Further supporting the role of phosphorylation in paracellular protein regulation, the results of a study examining brain endothelial cells exposed to ethanol suggest that the increased permeability of bovine brain microvascular endothelial cells (BBMEC) observed following ethanol exposure is linked to the phosphorylation of claudin-5 and occludin, likely mediated by myosin light chain kinase (MLCK) [150].

Acetaldehyde has also received considerable attention for its ability to alter the intestinal barrier function following alcohol consumption, and this subject has been thoroughly reviewed elsewhere [151]. Like ethanol, acetaldehyde was also shown to increase membrane permeability of an in vitro 3-D Caco-2 spheroid model exposed to acetaldehyde concentrations as low as 0.025 mM; yet, this exposure did not affect the mRNA levels of TJ proteins when this same model was exposed to 0.2 mM acetaldehyde [138]. Additionally, acetaldehyde also affects the localization of ZO-1 and causes an increase in global protein acetylation, including α-tubulin [138]. Indeed, considerable research has been devoted to determining the role of acetaldehyde in altering the localization of ZO-1 and other paracellular barrier proteins (for a comprehensive list, see [151]). As previously mentioned, various PTMs appear to play a significant role in determining the integrity of the barrier, whereby the presence of acetaldehyde leads to the persistent phosphorylation of several paracellular proteins, thus contributing to gut leakiness [140]. Furthermore, in vivo studies using WT and ALDH2(+/−) mice demonstrated increased intestinal permeability at the distal and proximal colon, jejunum, and ileum in ALDH2(+/−) mice following ethanol exposure, compared to the corresponding WT, which only displayed increased permeability at the distal colon [152]. This suggests an important role for acetaldehyde in mediating gut leakiness and for ALDH2 in moderating elevated acetaldehyde levels and thus affecting the re-distribution of TJ and AJ proteins in the mouse ileum and colon [152]. Interestingly, acetaldehyde was also shown to negatively impact an in vitro model of mucin-producing goblet cells, particularly through another catalyst of intestinal barrier dysfunction: oxidative stress [153].

Indeed, many adverse cellular manifestations, such as lipid peroxidation, and DNA damage arise as a result of oxidative stress, which ultimately leads to inflammation and apoptosis [47,74]. Oxidative stress was observed in the intestines of rats exposed to chronic alcohol conditions, evidenced by the increase in iNOS protein levels, nitrate and nitrite levels, and jejunal, ileal, and colonic protein nitration and oxidation, which correlated with increased intestinal permeability [98,154,155]. Specifically, the iNOS-dependent increase in miR-212 levels in Caco-2 cells exposed to ethanol appears to contribute to perturbations in the cell permeability [156]. In addition, an iNOS-mediated decrease in ZO-1 expression was observed in the ethanol-exposed Caco-2 cells [156]. The role of iNOS in instigating in-vitro increases in Caco-2 cell permeability was confirmed in-vivo using iNOS-KO mice, which displayed significantly less intestinal barrier dysfunction compared to the corresponding WT mice following exposure to alcohol in both mouse strains [156]. Additionally, elevated and activated iNOS appears to contribute to the activation of the Snail transcription factor, which was demonstrated to play a role in increased gut permeability [157]. However, iNOS is not the only enzyme responsible for the alcohol-induced development of oxidative stress. In a binge alcohol rat model, both iNOS and CYP2E1 protein levels were elevated in the intestines 1 or 2 h following the last binge dose [132]. CYP2E1 is able to oxidize ethanol into acetaldehyde, but in doing so, generates ROS molecules, such as the 1-hydroxyethyl radical [60,158]. Indeed, the role of intestinal CYP2E1 on gut leakiness observed following alcohol exposure has been well reviewed and involves important circadian proteins, such as CLOCK and PER2 [155]. Importantly, the observed increase in plasma endotoxin levels, oxidative stress (determined via increased 3-nitrotyrosine (3-NT) levels), and intestinal permeability was dependent on intestinal CYP2E1, since *Cyp2e1*-null mice were quite resistant to binge alcohol-mediated gut leakiness and subsequent fatty liver [120,132,159]. Additionally, the levels of nitrated and ubiquitin conjugated TJ and AJ proteins were markedly diminished in the same *Cyp2e1*-null mice exposed to binge alcohol, suggesting a role of CYP2E1 and/or CYP2E1-generated oxidative and nitrosative/nitrative stress in modulating the intestinal PTM landscape [120,132,159].

Furthermore, it is also possible that ROS can be provided through activated NADPH-oxidase isozyme(s) (NOXs) present in the colon and in immune cells in the lamina propria of the GI tract [32,160,161]. However, treatment of Caco-2 cells with acetaldehyde, ethanol, and the NADPH oxidase inhibitor diphenyleneiodonium did not ameliorate evidence of increased barrier permeability and dysfunction as *N*-acetyl cysteine was demonstrated to do [162]. The oxidative stress-induced mitogen-activated protein kinases (MAPKs) [e.g., p38 protein kinase (p38k), c-Jun *N*-terminal protein kinase (JNK), and extracellular signal regulated protein kinase (ERK)] are also implicated in gut barrier damage, since they are observed to be phosphorylated (activated) in Caco-2 cells and played a role in TJ disruption and gut barrier dysfunction, particularly through increasing MLCK mRNA [163], whose activity has been implicated in regulating intestinal permeability [164]. The oxidative stress-mediated decrease in hepatocyte nuclear factor-4α (HNF-4α) was also implicated in tight junction disruption following alcohol exposure [165]. Additionally, studies have implicated the fatty acid ethyl esters ethyl oleate and ethyl palmitate in increased ROS production in Caco-2 cells and these esters also caused redistribution of the paracellular proteins ZO-1 and occludin and caused increased ROS-mediated permeability of these cultured cells [166].

### 4.4. Crosstalk among Gut Dysbiosis, Intestinal Barrier Dysfunction, and ALD

Bidirectional gut–liver communication occurs as gut-derived molecules pass through the intestinal barrier and enter the portal vein to reach the liver, while liver-derived molecules pass through the biliary tract to interact with the intestines [20,167]. Typically, in a resting, non-disease state, SCFAs, secondary bile acids and other diet-derived metabolites will pass into the liver to perform various functions, such as driving fatty acid oxidation in the case of SCFAs [168], or, in the case of secondary bile acids, which have been modified in the gut, these metabolites can be recycled for future use [20]. Concurrently, the liver will secrete primary bile acids, among other liver-derived molecules, to the intestines to increase lipid absorption [169] and potentially alter the gut microbial landscape through their antimicrobial properties [170]. Heavy alcohol consumption will alter communication between the gut and liver, not only by increasing the presence of circulating alcohol [108] and subsequently produced acetaldehyde [171], which can damage the liver at high concentrations, but also by enabling harmful gut microbiota-derived molecules to leak into the circulation and inflict damage on both the intestinal barrier and the liver [167].

The non-luminal side of the intestinal barrier is home to the lamina propria containing a wide variety of immune cells, which can respond to changes in gut barrier dysfunction. The response of immune cells (specifically leukocytes) to alcohol consumption does not appear to be uniform across the lower GI tract [172], possibly due to the differences in microbial composition and abundance found throughout the GI tract, gut environmental factors, and/or differing ethanol concentrations at these regions. Predictably, following ethanol consumption, these cells and others will interact with and be activated by luminal-derived molecules (LPS/PAMPs) that traversed the damaged, leaky intestinal barrier [127]. LPS is a significant luminal-derived molecule that will enter circulation and directly alter local (gut) regions, triggering inflammation and injury to downstream organs, such as the liver, inflicting further damage [127,172] and enterocyte barrier dysfunction [173]. Although passage of LPS through the intestinal barrier has long been speculated to occur via paracellular transport, a recent study examining LPS transport across the intestinal barrier during lipid absorption (non-alcohol conditions) observed CD63- and lipid raft-mediated transcellular transport of LPS across the barrier and absorption into the portal vein [174]. Regardless of the transport method across the barrier, the manifestation of endotoxemia is a common occurrence in individuals with ALD [175,176] and serum LPS levels appear to correlate with the amount of alcohol being consumed [177]. However, before LPS and other gut-derived harmful molecules (ethanol, acetaldehyde, cytolysin, candidalysin [178], DAMPs, exosomes, etc.) can travel through the portal vein to the liver, the gut-localized immune response will be affected [174]. For example, an early study examining mice chronically exposed to alcohol demonstrated that the mRNA levels of certain cytokines (IL-1β and TNFα) are upregulated in the ileum by ethanol alone, while others (IL-6 and IL-11) are upregulated in the presence of both ethanol and LPS [179]. A more recent study using a binge on chronic mouse model of alcohol exposure found increased production of IL-17 from Paneth cells, which contributed to inflammasome activation in the small intestine, evidenced by increases in activated caspase-1 and IL-18 [134]. Furthermore, alcohol exposure has been shown to alter populations of immune cells in the gut, such as T-lymphocyte populations in the small intestine of Rhesus macaques chronically exposed to alcohol [180]. Importantly, considering that NADPH oxidase levels are increased during exposure of mice to the Western-style HFD [32,161] and play a role in inflammation of the intestines during dextran sodium sulfate (DSS)-mediated colitis [160], it is possible that ROS generated by NADPH-oxidase isozyme(s) present in immune cells in the GI tract may also contribute to gut-localized inflammation in response to alcohol.

LPS and other luminal-derived molecules will travel through the portal vein and reach the liver, where they will interact with various cell types and influence the different stages of ALD pathogenesis [181,182,183]. Briefly, depending on the specific DAMP and/or PAMP, gut-derived molecules will interact with specific toll-like receptor (TLR) complexes present in resident liver macrophages, Kupffer cells (and other liver cells) to initiate a specific inflammatory cascade [181,182,183]. In particular, the pathway by which LPS induces inflammatory cascades in the liver has been investigated thoroughly [184]. Briefly, after the binding of LPS to LPS-binding protein (LBP) and CD14, LPS will bind MD2 to interact with TLR4 on the membrane, thus activating the complex [181,182,184]. The resulting TLR4 activation can elicit intracellular signaling cascades which, depending on the specific proteins involved [182,184], can increase the expression of proinflammatory cytokines, such as TNFα, which will both increase apoptotic liver damage and hepatic inflammation [185,186]. Indeed, LPS has been shown to drive steatosis and inflammation through mechanisms such as decreased autophagic response in the liver [187] and increased pro-inflammatory cytokine production [188], respectively. Additionally, the LPS-mediated induction of the liver damage has also been implicated in increased hepatic stellate cell (HSC) response to TGF-β and fibrosis onset, although LPS may also exhibit anti-fibrotic properties through by targeting HSC proliferation, as recently reviewed [189]. Importantly, a key mediator of gut-induced damage of the liver is oxidative stress and, specifically, ROS, generated not only by LPS-activated Kupffer cells but via the oxidative metabolism of ethanol by CYP2E1 as high concentrations of ethanol enter the liver from the circulation [67].

A recent study demonstrated that increased ROS levels, likely generated by CYP2E1, occur alongside ethanol-induced decreases in autophagy in alcohol-exposed mice [187]. Furthermore, studies have confirmed the role of CYP2E1 in LPS-induced liver damage via the production of ROS and activation of oxidative stress-sensitive downstream kinases, such as the MAP kinases, and mitochondrial damage [190,191]. MAP kinases are not only activated by CYP2E1-generated ROS [68,79], but also by the NADPH oxidase-mediated production of ROS, which activate ERK in a CYP2E1-independent manner, leading to TNF-α production [192]. Another study found that arachidonic acid supplementation with alcohol activated ERK via ROS and, subsequently, increased TNF-α levels, supporting the notion that other factors, such as dietary n-6 fatty acids, can act alongside LPS in inducing oxidative stress-mediated liver damage following alcohol exposure [193] or during other liver pathologies, such as palmitic acid-mediated NAFLD [194]. In a binge alcohol model, NADPH oxidase-mediated ROS production is also important for inflammatory signaling by increasing interleukin-1 receptor-associated kinase (IRAK) levels in Kupffer cells 21 h post binge alcohol exposure, a change that was dependent on NF-κB activity and which correlated with increased TNF-α levels in these Kupffer cells [195]. Interestingly, at the PTM level of regulation, LPS and acetate (and/or acetaldehyde) were demonstrated to decrease hepatic SIRT1 levels, with, expectedly, increased hyperacetylation of a subunit of nuclear transcription factor kappa B (NF-κB), leading to increased inflammatory response in in vitro rat Kupffer cells [196]. Though not explicitly evaluated, the authors hypothesized that ROS may regulate this mechanism.

Kupffer cell-localized NADPH oxidase (NOX) was hypothesized early on to generate ROS during infiltration of LPS and/or neutrophils following alcohol exposure [197] and the role of specific members of the NOX family of NADPH oxidases, such as NOX4, has been described. Indeed, recent in vitro and in vivo models of alcohol-induced liver damage confirmed the role of NOX4 in increasing mitochondrial ROS and mitochondrial-mediated apoptosis, in addition to a partial role in steatosis development following alcohol exposure [198]. Translational approaches to combat macrophage-mediated ROS production during ALD have also revealed the mechanistic regulation of macrophage-localized NADPH oxidase, as globular adiponectin was shown to inhibit ROS and NOX2 expression through activation of liver kinase B1 (LKB1) and AMP-dependent protein kinase (AMPK) [199]. Besides immune cells, HSCs can also be targeted by LPS and, mechanistically, oxidative stress also appears to drive LPS-mediated increases in MCP-1 and IL-6 in HSCs [200]. Although gut-mediated hepatic oxidative stress represents an important contributing factor to ALD pathogenesis and progression, not all aspects of ALD pathogenesis appear to rely on oxidative stress, since other factors, such as insulin resistance, may also play a role [201].

## 5. The Antioxidant Properties, Metabolisms, and Health Benefits of Various Phytochemicals against Gut Dysbiosis, Intestinal Barrier Dysfunction, and Fatty Liver Disease

As described in the previous sections, gut dysbiosis with decreased ratios of Bacteroidetes/Firmicutes and changes in the levels of various endogenous metabolites, including ethanol and SCFAs, are associated with many disease states. Thus, there have been many efforts to normalize or restore the gut microbiome by using different diets, such as microbiota-targeted vegan (vegetarian) diets and/or consuming microbiota-accessible carbohydrates (oligocarbohydrates) or dietary supplements with various phytochemicals [19]. In addition, many phytochemicals represent diverse classes of antioxidants contained in a variety of fruits and vegetables as well as medicinal plants [202]. Most of these phytochemicals are known to have very low bioavailabilities due to their water insolubilities [202,203]. These phytochemicals include different chemical classes, such as various polyphenols, lignans such as phytoestrogens, carotenoids, phytosterols/phytostanols, alkaloids and glucosinolates, sulfur-containing compounds, etc., as recently reviewed [202]. In one class of phytochemical, there are many chemical subgroups. For instance, polyphenols represent antioxidant chemicals of many different subgroups. Common subgroups include flavonoids (e.g., apigenin, quercetin) polyphenolic amides (e.g., capsaicin) and polyphenolic acids [204], which include important polyphenols such as resveratrol, curcumin, capsaicin, quercetin, rutin, genistein, daidzein, ellagic acid, and proanthocyanidins (tannoids), such as (-)-epicatechin (EC), (-)-epigallocatechin (EGC), (-)-epicatechin-3-gallate (ECG), and (-)-epigallate catechin-3-gallate (EGCG), among others, as recently reviewed [205]. Despite the different chemical structures of these phytochemicals, they have exhibited their beneficial effects in some human studies, many experimental rodent models, and in vitro cell culture models.

One of the characteristic properties of these phytochemicals is their very low water solubility, despite the presence of many hydroxyl groups in polyphenols [206,207]. It is known that some phytochemicals can be metabolized or modified by intestinal bacteria (specifically hydrolyzed, reduced, deglycosylated, degraded, conjugated, etc.) and some of these alterations may enhance absorption of these chemicals in the intestine [208,209,210]. For instance, resveratrol, contained in grapes, berries, and red wine, can be metabolized to dihydroxyresveratrol, 3,4′dihydroxybibenzyl (lunularin), and 3,4′-dihydroxy-trans-stilbene by gut bacteria or converted to the more bioavailable metabolite piceid [205,211]. Likewise, 3,4-dihydroxyphenylacetic acid, a gut-derived metabolite of the flavonoid quercetin, found that various fruits, vegetables, and beverages (mostly as quercetin glycosides) [212] can both act as a free radical scavenger and reduce markers of advanced glycation end products [194]. A multitude of polyphenols (e.g., flavonoids, thearubigins, chlorogenic acids) are known to be metabolized by gut microbiota [213] as well as conjugated, glucuronated, deglycosylated, etc., in the intestinal epithelial cells and liver, although each compound is supposed to produce its own unique structural derivatives. As reviewed elsewhere, these chemicals will undergo gut bacteria-mediated metabolic transformations or conjugation, which may increase absorption, as in the case of anthocyanins [214] and phytoestrogens [215]. In addition, bacterial transformations may yield new metabolites with beneficial effects, as in the case of equol derived from daidzein [215], where equol is known to possess anti-inflammatory and anticancer properties [214]; however, not all microbial-derived metabolites exhibit beneficial effects or increased absorption [214,215]. In addition, many phytochemicals, despite their low bioavailabilities, are known to exert their biological activities in the gut independent of tissue absorption. In fact, these phytochemicals are likely to be present in the highest abundance in the gut after oral consumption, mainly due to very low absorption rates through the gut membrane [208,209,210]. These phytochemicals exhibit their beneficial effects by altering the rates of transcription of certain genes and the absorption or levels of many essential nutrients or compounds, such as cholesterol, triglycerides, and bile acids [208,209,210]. Furthermore, they are known to exert their functional activities by their unique antioxidant activities by suppressing the activities of pro-oxidant enzymes and transcription factors, including NF-kB, which transcriptionally regulates many proinflammatory downstream targets such as TNFα and iNOS. These antioxidant compounds also show anti-inflammatory effects with decreased levels of pro-inflammatory cytokines and chemokines against gut-related disorders such as inflammatory bowel disease [216], in addition to combatting other metabolic syndrome-related symptoms (e.g., obesity-related fatty liver and type 2 diabetes associated cardiovascular disorders) [210] and possibly preventing neurodegenerative abnormalities, including Alzheimer’s disease, through the gut(–liver)–brain axis [217].

As illustrated in Figure 2, oxidative and nitrosative/nitrative stress, inflammation and leaky gut, endotoxemia and inflammatory tissue injury arise following exposure to many environmental compounds such as ethanol (alcohol drinking), Western-style high-fat diets, and high fructose or genetic risk factors and/or certain disease states. However, certain phytochemicals and/or their metabolites are capable of ameliorating these pathological manifestations caused by some of these insults, particularly through changing (usually normalizing) the amounts and components of the gut microbiota, resulting in improved composition (i.e., gut eubiosis) [202,218,219,220] from gut dysbiosis. Additionally, these beneficial agents can alter the metabolism and production of many endogenous compounds such as ethanol, acetaldehyde, SCFAs, bile acids, and trimethylamine (TMA) by altering the compositions of the gut microbiome [202,221] or perhaps even ethanol and acetaldehyde, which have been shown to be produced by some gut bacterial strains [202,218,219,220,222]. These antioxidant phytochemicals and/or their gut metabolites can also suppress the increased oxidative and nitrosative/nitrative stress by inhibiting the enzymes, such as CYP2E1, NOXs, iNOS, and NF-κB [223,224,225], as listed in the Figure. Furthermore, these beneficial antioxidant compounds from dietary supplements can affect key enzymes/proteins in the cellular signaling pathways, such as AMPK [226] and hepatic Sirt-1 [227], or transcription factors, including NF-κB and PGC-1α [228,229], to show their biological effects, as demonstrated by polyphenols in green and black teas [230]. In addition, these antioxidant phytochemicals may improve the disease states by modifying the genetic and epigenetic regulations in the gut and other tissues possibly through the NAD^+^-dependent non-histone deacetylase Sirt-1 [227] and its isoforms. Finally, gut dysbiosis-mediated altered levels of SCFAs, especially propionate and butyrate, which are inhibitors of histone deacetylase [19], may result in different epigenetic profiles, which further regulate the gene transcription rates compared to those during eubiosis and/or in phytochemical-treated cases. All these changes are likely to contribute to the beneficial effects of phytochemicals and improvement of various disease states in experimental models, as well as in a few human studies.

## 6. Translational Approaches and Therapeutics against Alcohol-Mediated Oxidative Stress, Gut Dysbiosis, Intestinal Barrier Dysfunction and Fatty Liver Disease

The presence of pathogenic microbes and their metabolites in the gut may potentially promote ALD and its progression, since pretreatment with non-absorbable antibiotics such as neomycin and polymyxin B [231] or rifaximin [232] significantly prevented the severity of alcohol-mediated liver disease in different animal models. Yet, although neomycin and polymyxin B helped ameliorate fructose-mediated steatosis development [233], rifaximin only showed limited success in combatting certain manifestations of non-alcoholic steatosis and steatohepatitis (NAFLD/NASH) [234] and thioacetamide-induced liver injury [235]. Furthermore, gut microbiome analyses of control and age-matched cirrhosis patients revealed that altered gut microbiota is also positively associated with cirrhosis and progression [236], suggesting an important role of gut dysbiosis in various liver diseases. The opposite case may also be true, where the absence of all bacteria is also liable to drive liver damage, since one study showed that germ-free mice exposed to binge alcohol conditions displayed greater hepatic fat accumulation and inflammation compared to WT mice, in addition to increased CYP2E1 mRNA in the proximal small intestine, which may trend toward elevated CYP2E1 protein levels and, likely, a stimulation of ROS production and intestinal and liver damage [237]. This germ-free study and reports characterizing gut dysbiosis following alcohol exposure demonstrate that while the gut dysbiosis has the potential to escalate ALD, the complete absence of gut microbes does not necessarily attenuate manifestations of ALD. Thus, maintenance of the normal balance of microbes should help prevent the initiation of the ‘second hit’ in the two-hit hypothesis of ALD progression, whereby early, mild manifestations of ALD (fatty liver) can progress to more severe insults (inflammation or fibrosis) through a second hit, such as certain gut-derived molecules, such as LPS and reactive oxygen species, as proposed in NAFLD/NASH [238,239,240,241]. Indeed, numerous therapeutic studies have sought approaches to moderate oxidative and nitrosative/nitrative stress, gut dysbiosis, and intestinal barrier dysfunction following alcohol exposure. These approaches include supplementation with commensal microbes, gut-protective factors, and a wide range of dietary options from both synthetic and natural origins.

Numerous studies have reported the beneficial effects of commensal microbial (probiotics) supplementation on ALD manifestations and gut dysbiosis [242]. Recent exciting results with clinical studies revealed that fecal microbiota transplantation (FMT) from healthy donor people seems to be relatively safe and effective in attenuating the severity with improvement of health outcome in mouse models of alcoholic liver injury [243] and patients with severe alcoholic hepatitis or cirrhosis [244,245,246], as reviewed [85]. In addition to whole FMT, supplementation of a specific microbial strain(s) such as lactobacillus [247] can also improve systemic endotoxemia and liver conditions following alcohol consumption. Additional evidence for the beneficial role of microbial strain supplementation includes the decrease in high blood ALT and AST levels following ethanol exposure to rats supplemented with yogurt or cream cheese made with *Lactococcus chungangensis* CAU 28 [248], the decline in hepatic iNOS and global nitration levels following *Lactobacillus fermentum* administration, depending on time point [249], and the *Roseburia intestinalis*-mediated reduction in gut permeability through increased occludin mRNA and protein levels [250]. Additionally, other members of the Lactobacillus genus such as *Lactobacillus plantarum*, have been shown to reduce inflammatory markers, triglyceride levels and gut leakiness associated with ALD by way of epidermal growth factor receptor (EGFR) activity [251]. Inflammation, steatosis, and gut leakiness were also mitigated in a chronic alcohol mouse model supplemented with *Akkermansia muciniphila* gut population [106], which, as previously stated, is found to be decreased in individuals with steatohepatitis [106] and alcohol use disorder [20,102].

Alongside probiotics, certain dietary supplements have been shown to attenuate ALD pathogenesis and limit gut dysbiosis and leakiness. For example, administration of the comestible cricket *Gryllus bimaculatus* prior to acute alcohol exposure was demonstrated to mitigate ethanol-induced increases in intestinal oxidative stress (8-OHdG levels), hepatic apoptotic markers, and hepatic triglyceride accumulation [252]. Pomegranate extracts and indole-3-carbinol (I3C), derived from Brassica vegetables, both mitigated inflammation of the liver and hepatocyte and enterocyte apoptosis in binge alcohol [120] and chronic plus binge alcohol exposure models, respectively [159]; although, unlike pomegranate [120], I3C supplementation did not attenuate fatty liver [159]. Additionally, ellagic acid (EA) and urolithin A (UA), two polyphenols derived from pomegranate extracts, mitigated ethanol-induced increases in gut permeability in T84 colon cells [120]. Furthermore, EA, UA, and I3C were all capable of reducing ethanol-induced increases in hepatic CYP2E1 levels, in vivo (I3C) [120,159] and in vitro (EA, UA) [120], which should reduce hepatic oxidative stress and attenuate liver injury. Interestingly, the PTM landscape is also affected by these supplements, whereby both I3C and pomegranate supplementation decrease ethanol-induced hyperacetylation in the liver and intestines, respectively [120,159]. Pomegranate extract, in particular, reduced ethanol-induced global nitration and ubiquitination of intestinal proteins and, specifically, ethanol-induced claudin-1 nitration and ubiquitination, thus preventing its degradation, and all of these PTM changes correlated with the prevention of intestinal barrier damage, and endotoxemia [120]. Additionally, rats chronically administered alcohol and fed oats exhibited decreases in ethanol-induced protein nitration and oxidation in all small intestine subregions and the colon, which was suggested to be due to a decrease in oxidative stress as a result of oat-mediated decreases in iNOS, nitrite, and nitrate levels [253]. Additionally, β-glucans from various foods, such as oats and barley [224], were effective against gut dysbiosis in non-alcohol-related diseases, thus supporting human health [225]. When co-administered with ethanol at various percentages, fish oil (high in n-3 polyunsaturated fatty acids) was shown to attenuate liver manifestations of ALD (steatosis and inflammation) [90], plasma endotoxin levels [90], alcohol-induced intestinal permeability dysfunction [254] and even impacted gut microbial composition, through a recovery of fecal Bifidobacterium members [254] and an increase in Bacteroidetes members (especially with supplementation of 25% fish oil), thus decreasing the ratio of Firmicutes to Bacteroidetes following alcohol exposure [90]. However, studies have reported that oxidation of fatty acids in fish oil (prior to administration) actually worsens liver outcomes following alcohol exposure and even increases the abundance of members of the Gram-negative, LPS-producing Proteobacteria phylum [255]. Thus, proper maintenance of fish oil in non-oxidized states appears to be very important to ensure its beneficial effects against alcohol-mediated fatty liver injury [256], possibly through preventing leaky gut [254].

Supplementation of zinc [130] may also attenuate liver and gut dysfunction following alcohol exposure. Zinc levels were observed to decrease in the ileum of mice chronically exposed to alcohol, which correlated with plasma endotoxemia, ileal oxidative stress and permeability, and decreased tight junction protein levels [257]. Mechanistically, by hampering the activity of enzymatic regulators of hepatic apoptosis (e.g., caspase-3), zinc supplementation can prevent hepatocyte death following alcohol exposure [258] and may benefit from, but does not require, the zinc-binding ability of metallothionein to exert its therapeutic effect [259].

Some alternative and complimentary remedies, such as traditional herbal medicines in China [260,261,262], Korea [263], and India [264,265], have recently proved effective in both combatting inflammation and in altering the gut microbial composition in models of ALD or models of intestinal disease. For example, co-administration of ethanol with the fungi *Wolfporia cocos* (or, more specifically, the water-insoluble polysaccharides from their fruiting bodies) decreased the hepatic triglyceride levels and MCP-1 levels, indicating decreased steatosis and inflammation of the liver [261]. Interestingly, these polysaccharides increased Firmicutes abundance and decreased the abundance of the Gram-negative, LPS-producing Proteobacteria phylum following chronic alcohol exposure [261]. Additionally, leaf extract from the plant *Dendropanax morbifera* leaf extract also altered microbial composition in rats acutely exposed to alcohol, whereby, for example, members of the Bacteroides operational taxonomic unit (OTU) increased upon co-administration of leaf extract with ethanol, compared to the ethanol group [263]. The *Dendropanax morbifera* leaf extract co-administered with ethanol contained several phytochemicals (e.g., rutin, caffeic acid, etc.) [263] and it is highly likely that the list of beneficial antioxidant phytochemicals capable of protecting against alcohol-mediated oxidative stress, gut dysbiosis, epithelial barrier dysfunction, and ALD (as well as NAFLD) will be increased in the future.

Microbiota-derived molecules may also help combat alcohol-induced tissue damage and ALD pathogenesis. Indeed, while microbial products derived from the colon and feces of alcohol-exposed mice were demonstrated to increase intestinal permeability in vitro, in addition to instigating in vitro T-cell activation [266], certain beneficial microbial products may reverse these adverse outcomes. SCFAs, in particular, butyric acid supplemented as tributyrin, may improve intestinal barrier functioning through the preservation of ZO-1 and occludin protein levels and proper localization at the barrier [267]. Additionally, butyrate can restrain cytokine (TNFα and MCP-1) production following acute alcohol exposure, although it does not appear to ameliorate alcohol-induced steatosis [267]. Indirect benefits of SCFA supplementation might also be achieved through administration of *Pediococcus pentosaceus*, which increased the levels of the SCFAs possibly due to the partial recovery of bacterial genera, such as Clostridium, whose levels were decreased in the mouse model of chronic plus binge alcohol exposure and whose recovery could increase SCFA production, as suggested by the positive correlation found between Clostridium and butyric acid levels in this study [268]. Administration of other SCFAs or other beneficial microbial products may yield similar results to butyrate supplementation, such as administration of indole-3-propionic acid (IPA), an indole metabolite generated by the metabolism of tryptophan, which elevated gut tight junction proteins, thus preventing intestinal barrier dysfunction and liver damage in rats exposed to a Western-style HFD, possibly by the gut–liver axis [269]. However, another study exposing mice to HFD did not find a reduction in liver triglyceride levels with co-administration of IPA (in addition to other parameters of bodily damage or inflammation), which could be attributed to minor differences in the HFD or murine model used [270]. Nevertheless, these differing results highlight the need for continued studies on the role of microbial products in potentially ameliorating liver and tissue damage induced by a wide range of stimuli.

Certain synthetic drugs, such as sennoside A [271] and metformin [272,273], can exhibit their beneficial effects on leaky gut and endotoxemia, in addition to preventing NAFLD and/or learning and memory impairment. However, it is also known that some antibiotics and chemotherapeutic agents such as 5-fluouracil and vancomycin are known to cause dysbiosis of gut microbiota [274]. In the latter cases, a probiotic with digestive enzymes was formulated to protect against cancer-drug-related dysbiosis, suggesting a reminder of careful interpretation of the results in using some drugs to regulate gut dysbiosis. As mentioned in the previous sections, usage of some antibiotics such as neomycin and polymyxin B or rifaximin could be considered for treating alcohol-mediated gut endotoxemia and steatotic and inflammatory liver damage through repurposing their applications, since these beneficial changes were observed in rodent models [231,232]. Additionally, the composition of the gut microbiota was changed in several orders (e.g., Erysipelotrichales) following ethanol and ethanol with rifaximin administration; however, the physiological role of these compositional changes and others should be further studied [232]. These antibiotics may represent possible options for treating gut dysbiosis, but future studies will have to confirm this proposition. However, aside from already-approved antibiotics, the usage of other synthetic compounds may not be practical, since large-scale randomized clinical studies are likely to require considerable time, cost, and effort in evaluating the safety and efficacy tests needed for FDA approval.

The benefits of naturally occurring compounds found in many edible plants, fruits, vegetables, and dietary supplements should also be considered as alternative approaches. For instance, various phytochemicals (berberine, curcumin, resveratrol, and the numerous components of silymarin) from different foods and plants can prevent NAFLD and, in some cases, can normalize gut dysbiosis and intestinal permeability changes, especially in rodent models [262]. The antioxidant *N*-acetylcysteine (NAC) was shown to mitigate LPS-induced intestinal permeability in the IPEC-J2 intestinal porcine enterocyte cell line [275], and experiments using Caco-2 cells showing that NAC supplementation with alcohol can prevent an EtOH-induced increase in CLOCK and PER2 expression helped formulate the hypothesis that ROS generated by CYP2E1 in response to ethanol could increase the expression of these circadian rhythm proteins, leading to intestinal dysfunction and permeability, as recently reviewed [276]. Some of these phytochemicals, such as resveratrol, curcumin, silymarin, genistein, quercetin, rutin, anthocyanidins, ellagic acid, etc., may have very limited bioavailability, although some may be better absorbed than others [277]. In theory, intakes of extremely large amounts of these natural compounds are needed to observe their beneficial effects due to their very low bioavailability, though experiments testing this hypothesis using curcumin or ellagic acid suggest this may is not the case [205,208,209,210,226,278]. These phytochemical compounds may have paradoxically high functional activity despite the setback of their low bioavailability [205]. For instance, a subset of French people, who usually consume a diet with relatively high saturated fatty acids and cholesterol, exhibit low risks of major diseases, like coronary heart disease, and deaths known as the French paradox [279,280] possibly due to their habits of daily drinking of wine, which contains resveratrol and other antioxidants [281]. Part of the reason could be that these polyphenol compounds may have direct antioxidant effects that counteract potentially oxidative enzymes such as NADPH oxidases [282] and CYP2E1 [223,283]. Alternatively, these chemicals may possibly exhibit their benefits by preserving the activities and/or levels of certain antioxidizing enzymes and proteins, as suggested by the importance of the presence of SOD2 for the resveratrol-mediated prevention of cytotoxicity in mouse hippocampal neurons pretreated with this polyphenol [284]. Additionally, other antioxidant-related proteins, such as ALDH2 and glutathione peroxidase (Gpx), which were demonstrated to be inactivated in the presence of acetaminophen [285], alcohol or non-alcoholic substances [68,79], may also be restored by the administration of antioxidant phytochemicals. Furthermore, these phytochemicals, including various flavonoid compounds, demonstrate their benefits by preventing gut dysbiosis and normalizing the amount and composition of beneficial bacteria such as *Lactobacillus, Akkermansia,* and *Bifidobacteria*, while decreasing the population of potentially harmful microbes, as suggested elsewhere [219,262,286]. In addition to their known beneficial interaction with the gut microbiota [202], certain phytochemicals may also affect bacterial strains known to produce ethanol and acetaldehyde, both of which are known to cause leaky gut and fatty liver disease [120,121], although this hypothesis has yet to be tested. In general, the usage of these phytochemicals and/or other prebiotics or probiotics as dietary supplements could be important in treating patients with alcoholic steatohepatitis with leaky gut and endotoxemia, especially considering the increasing incidence of ALD and the proposed high mortality rate (30~40%) at 1 month for individuals with severe alcoholic hepatitis [287,288,289]. Unfortunately, there is no clinically proven drug approved for effectively treating alcoholic hepatitis patients, although a few clinical studies are being conducted [290,291]. Early clinical trials using FMT from healthy donors into patients with alcoholic hepatitis, cirrhosis, AUD, etc., offer preliminary data suggesting that FMT may be a safe and promising therapeutic option [85,244,245,246], although additional large-scale and long-term multi-center studies are needed to confirm the safety/toxicity profile and efficacy of the newly emerging FMT therapy against various liver diseases, including ALD. Requests have been submitted for several phytochemicals listed in this review, such as trans-resveratrol, urolithin A, and curcumin, to be recognized as generally regarded as safe (GRAS by the FDA definition), and, if given FDA approval, these could be recommended as dietary supplements for ALD patients in addition to their clinical treatment protocols. However, these newly emerging areas need to be further studied to completely understand the benefits of various antioxidants contained in fruits, vegetables and dietary supplements.

## 7. Conclusions

In this review, we have briefly described the role of gut microbiota alteration during normal development as well as in pathological conditions. We also mentioned the role of gut dysbiosis caused by various environmental factors as well as epigenetic and/or genetic risks. We also describe the patterns of gut microbiome changes in various pathophysiological conditions and provide evidence for the benefits of ameliorating gut dysbiosis by many different agents in human populations and experimental rodents as well as cell culture models. Moreover, we describe the causal role of increased oxidative stress in promoting intestinal barrier dysfunction, subsequently leading to elevated endotoxemia and fatty liver disease. We also described potentially safe methods of translational approaches against alcohol-mediated oxidative stress, gut dysbiosis, leaky gut, and ALD. These beneficial methods and agents include healthy lifestyle changes with proper intake of healthy diets, containing beneficial chemicals contained in many fruits and vegetables, or n-3 polyunsaturated fatty acids. Indeed, these beneficial agents have demonstrated their effectiveness against gut dysbiosis, intestinal barrier dysfunction, endotoxemia, and ALD and NAFLD in many experimental models. In particular, supplementation with certain phytochemicals represents an exciting prospective therapeutic option for mitigating alcohol-mediated tissue damage owing to the breadth of known phytochemicals and the observed benefits of these chemicals and/or their metabolites on the gut microbiome and redox regulation, despite many compounds having low bioavailabilities. Importantly, future certification of many phytochemicals in various fruits, vegetables, and plants as GRAS could permit usage of these chemicals as dietary supplements in treating ALD. However, randomized large-scale clinical studies need to be conducted in the future to accurately demonstrate the efficacy of some of these antioxidant phytochemicals.

## Figures and Tables

**Figure 1 antioxidants-10-00384-f001:**
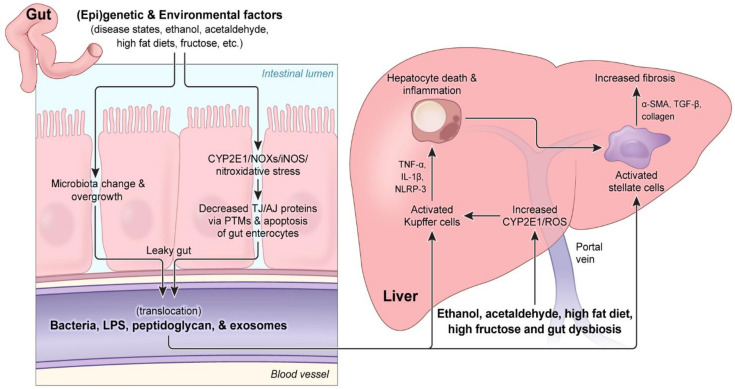
Schematic overview of gut–liver communication and damage prompted by intestinal disorders or consumption of exogenous agents. Numerous exogenous agents (e.g., alcohol, high-fat diet (HFD), fructose, etc.) or underlying intestinal disorders (e.g., Crohn’s disease, ulcerative colitis, etc.) can elicit changes to the abundance and/or composition of the gut microbiota. Elevated gut CYP2E1 and NADPH oxidases (NOXs) can increase oxidative stress. The resulting gut dysbiosis alters gut metabolism and damages the intestinal barrier through various mechanisms, including the oxidative stress-mediated post-translational modifications (PTMs), leading to decreases in paracellular junction complex proteins. Sustained damage to the barrier causes gut leakiness and, subsequently, a gut-localized immune response and increased levels of harmful gut-derived compounds (e.g., lipopolysaccharide (LPS), peptidoglycan, exosomes, etc.) into the circulation. LPS (and other gut-derived metabolites) and alcohol will reach the liver and drive alcoholic liver disease (ALD) pathogenesis and progression. Kupffer cell activation, mediated by LPS and oxidative stress driven by metabolism of the ethanol by hepatic CYP2E1 and from activated NOXs, increases inflammatory cytokine levels (e.g., TNF-α, etc.) and instigates hepatocyte apoptosis. Eventually, sustained oxidative stress, LPS infiltration, and hepatocyte damage will lead to the activation of hepatic stellate cells, driving liver fibrosis and continued liver damage.

**Figure 2 antioxidants-10-00384-f002:**
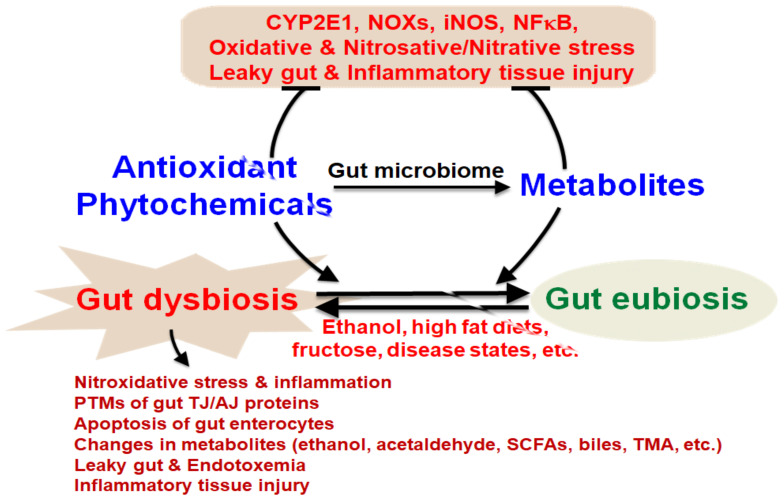
Proposed mechanisms of the beneficial effects of antioxidant phytochemicals on gut dysbiosis and oxidative stress-mediated intestinal barrier dysfunction and inflammatory liver injury. As described in the text, many phytochemicals contained in various fruits, vegetables and dietary supplements can be metabolized by gut microbiota for improved absorption, leading to greater bioavailability. By improving the oxidative stress and gut dysbiosis, these antioxidant phytochemicals and/or their metabolites prevent leaky gut, endotoxemia, inflammation, and alcoholic and/or non-alcoholic fatty liver disease.

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
