# Peer review of "Translational Approaches with Antioxidant Phytochemicals against Alcohol-Mediated Oxidative Stress, Gut Dysbiosis, Intestinal Barrier Dysfunction, and Fatty Liver Disease"

_antioxidants, 2021, doi:10.3390/antiox10030384_

Round 1
Reviewer 1 Report
The manuscript submitted to the MDPI Antioxidants journal as an article is of very high quality. It represents a comprehensive and systematic overview of the current state of the art related to the potential health benefits of the biologically active compounds/phytochemicals possessing antioxidative properties. The main topic of the submitted article is timely and highly relevant since it is related to the early signs of liver disfunction due to the prolonged presence of harmful substances such as ethanol and associated metabolites. This condition is quite often underestimated, and there are not readily available approaches for its diagnosis. The authors have combined clearly and objectively recent findings obtained through the studies of dysbiosis in the gut and production of the small metabolites by the intestine's microbiota. This approach is much appreciated and needed. Based on these observations and conclusions, the interested stakeholders (fundamental, clinical scientists, and GPs) would obtain insights into the events happening before the extensive liver damages, which is of significant value to the clinicians and the patients.
The figures associated with the text are of very good quality, comprehensive, and easily understandable at the same time.
I highly recommend publishing this manuscript in the Antioxidant Journal in the present form.
Author Response
Response: We have greatly appreciated the reviewer’s time and effort in reading our manuscript as well as the positive comments. To improve our manuscript, we have also revised other parts of the text a little bit.
Reviewer 2 Report
In general, the subject of this manuscript is well driven. The information of this review is relevant and is very useful for the scientific community, however minor revision needs before to be published.
The mechanisms of gut dysbiosis, leaky gut, endotoxemia, and fatty liver disease have been described not briefly as authors mentioned in the abstract. They have extensively described.
The Introduction section would be deleted and incorporated in the following section: The gut microbiome
The reviewer suggests to delete the word “Review” from the section 3 just Oxidative Alcohol Metabolism and Progression to Alcoholic Liver Disease
This section would be improved including a diagram showing the main reactions that take place during ethanol intake.
The section 4 related to the mechanisms of alcohol-mediated gut dysbiosis, intestinal barrier dysfunction, and consequences is excessively extended (9 pages). The authors may prepare a resuming table to make the text more comprehensible revealing the important trends to consider for ALD and gut dysbiosis prevention.
Figure 1 is clear and describes an overview of gut-liver communication and damage prompted by intestinal disorders or consumption of exogenous agents. It is missed “n” of environmental.
However, the sections 5 and 6 might be stronger, taking into account the title of manuscript. The reviewer suggests to include a figure representative with the phytochemicals in different plant foods and their metabolic transformations throughout the GI tract (chemical structure of phytochemicals, examples of food sources indicating an average concentration in each plant food matrix, .....)
Figure 2 needs to be improved its edition. The figure legend is not necessary is already included in the section 5.
In this sense, a new table is required to summarize numerous studies reported the beneficial effects of certain dietary supplements to attenuate ALD pathogenesis and limit gut dysbiosis and leakiness. The inclusion of this table would make the manuscript more clear and easier to read and understand the manuscript.
In the conclusions section, the authors describe the different sections of the review but they should state clearly the importance of supplementation with phytochemicals because of their health benefits against gut dysbiosis, intestinal barrier dysfunction, and fatty liver disease. The reviewer indicates the convenience of avoiding references in conclusion section that could be included in the previous sections.
Author Response
In general, the subject of this manuscript is well driven. The information of this review is relevant and is very useful for the scientific community, however minor revision needs before to be published.
The mechanisms of gut dysbiosis, leaky gut, endotoxemia, and fatty liver disease have been described not briefly as authors mentioned in the abstract. They have extensively described.
Response: We have greatly appreciated the reviewer’s time and effort in reading our manuscript as well as the positive comments. We understand the reviewer’s specific point. Thus we removed the word ‘briefly’ in the Abstract.
The Introduction section would be deleted and incorporated in the following section: The gut microbiome.
Response: We understand the reviewer’s specific point. However, we feel that it may be better to have an Introduction, although we added a few sentences at the end of the Introduction. In addition, the other two reviewers did not recommend the removal of the Introduction. Thus, we would like to keep it as in the original version.
The reviewer suggests to delete the word “Review” from the section 3 just Oxidative Alcohol Metabolism and Progression to Alcoholic Liver Disease.
Response: We appreciate the reviewer’s suggestion. Thus we removed the words ‘Review of’ in the Section 3.
This section would be improved including a diagram showing the main reactions that take place during ethanol intake.
Response: We understand the comment. However, the oxidative and non-oxidative pathways of ethanol metabolism are well-established and described in many review articles, as shown in references [45, 53, 77, 104, etc.]. Thus, to save space, we just mention them in the text without the actual diagram, as it was. We hope that our response is acceptable to this reviewer.
The section 4 related to the mechanisms of alcohol-mediated gut dysbiosis, intestinal barrier dysfunction, and consequences is excessively extended (9 pages). The authors may prepare a resuming table to make the text more comprehensible revealing the important trends to consider for ALD and gut dysbiosis prevention.
Response: We fully understand the reviewer’s comment. However, there are so many naturally-occurring compounds that are beneficial against alcohol-mediated gut dysbiosis and intestinal barrier dysfunction. Inclusion of a Table with extensive list of beneficial compounds seems too much and beyond the scope of this review. In addition, the other two reviewers did not ask the extensive changes. We also hope that our response is acceptable to this reviewer.
Figure 1 is clear and describes an overview of gut-liver communication and damage prompted by intestinal disorders or consumption of exogenous agents. It is missed “n” of environmental.
Response: We greatly appreciate the reviewer’s catching of our mistake. We corrected our mistake in the revised Figure 1.
However, the sections 5 and 6 might be stronger, taking into account the title of manuscript. The reviewer suggests to include a figure representative with the phytochemicals in different plant foods and their metabolic transformations throughout the GI tract (chemical structure of phytochemicals, examples of food sources indicating an average concentration in each plant food matrix, .....).
Response: We fully understand the reviewer’s suggestion. However, the new tasks to include many plant compounds and their transformation as well as their concentrations are beyond of the scope of this review. In fact, this issue has been described in many reviewed references #79 and 195-200. We hope that our response is acceptable to the reviewer.
Figure 2 needs to be improved its edition. The figure legend is not necessary is already included in the section 5.
Response: We generally agree with the reviewer’s suggestion. Thus, we shortened the legend for Figure 2.
In this sense, a new table is required to summarize numerous studies reported the beneficial effects of certain dietary supplements to attenuate ALD pathogenesis and limit gut dysbiosis and leakiness. The inclusion of this table would make the manuscript more clear and easier to read and understand the manuscript.
Response: We fully understand the reviewer’s suggestion. However, the new task to include certain dietary supplements to attenuate ALD pathogenesis via gut dysbiosis and leakiness is beyond of the scope of this review. In fact, this issue has been described in the references 79 and other references. We hope that this response is also acceptable to the reviewer. If not, please let us know.
In the conclusions section, the authors describe the different sections of the review but they should state clearly the importance of supplementation with phytochemicals because of their health benefits against gut dysbiosis, intestinal barrier dysfunction, and fatty liver disease. The reviewer indicates the convenience of avoiding references in conclusion section that could be included in the previous sections.
Response: We generally agree with the reviewer’s suggestion. Thus, we relocated some sentences with references into a previous section to avoid references in the Conclusion. We hope that our responses are satisfactory to this Reviewer.
Reviewer 3 Report
The Authors should describe the serum levels of endotoxin found in NAFLD. In addition they should analyze the molecular mechanisms by which endotoxin can induce the oxidant stress in NAFLD. Finally they should discuss the emerging role of increased oxidant stress as the link between endotoxemia and increased cardiovascular risk in NAFLD (Endotoxin found in the atherosclerosis plaque).
Author Response
The Authors should describe the serum levels of endotoxin found in NAFLD
Response: The serum LPS levels in NAFLD people and controls are 10.6 and 3.9 (EU/mL), respectively (Harte A et al, J of Inflammation 2010; 7:15 – PMID: 20353583). However, the LPS levels in rodents seem to fluctuate too much, depending on the different laboratories. For instance, Zhou et al reported that the LPS levels in HFD-fed mice and control were 27 and 22 (EU/mL), respectively (Zhou D et al World J Gastroenterol 2017;23:60-75 – PMID: 28104981). In contrast, LPS levels in fast food exposed mice and control mice were 0.7 vs 0.1 EU/mL, respectively. (Abdelmegeed MA et al, Sci Rep 2017;7:39764 – PMID: 28051126) while its levels in fast food and fructose exposed C57BL/6 mice were 0.034 versus 0.013 EU/mL in controls (Brandt A et al, Sci Rep 2019;9:6668 – PMID: 31040374). These results suggesting different levels in various labs possibly depending on the assay methods and experimental conditions.
In addition they should analyze the molecular mechanisms by which endotoxin can induce the oxidant stress in NAFLD
Response: Endotoxin (LPS), elevated through gut dysbiosis and leaky gut, can activate a pro-inflammatory transcription factor NF-kB (Ding L-Y et al, Oral Dis 2019; 25:178901797 – PMID: 31283861), which increases iNOS expression and proinflammatory cytokines, including TNFα, that further activate protein nitration (Yoo SH et al., Toxicol Lett 2011; 202:23-29 PMID: 21262334). In addition, LPS can activate p-JNK pathway (Ding L-Y et al, Oral Dis 2019; 25:178901797 – PMID: 31283861). Activation of iNOS and JNK can stimulate post-translational modifications and inactivation of various antioxidant proteins and mitochondrial dysfunction, leading to increased oxidative and nitrate stress (Li F et al, Food Chem Toxicol 2018; 120:104-111 – PMID: 29803697; Song BJ et al, Redox Biol 2014; 3: 109-123 – PMID 25465468). However, in this review, we have not discussed the mechanisms of LPS-mediated oxidative stress in NAFLD, since we mainly focused on the mechanisms of alcohol-mediated oxidative stress, gut dysbiosis, leaky gut and fatty liver disease as well as translational approaches of using antioxidant phytochemicals.
Finally they should discuss the emerging role of increased oxidant stress as the link between endotoxemia and increased cardiovascular risk in NAFLD (Endotoxin found in the atherosclerosis plaque).
Response: We fully understand the reviewer’s comment on the role of oxidative stress as the link between endotoxemia and increased cardiovascular risk in NAFLD, as reported (Li F et al, Oxid Med Cell Longev 2017;2017:2302896 – PMID: 28828145). As responded in the previous comment, elevated endotoxin can cause mitochondrial dysfunction with increased oxidative stress (Li F et al, Food Chem Toxicol 2018; 120:104-111 – PMID: 29803697). However, we only focused on alcohol-mediated gut and liver injury but not on cardiovascular damage, although we believe that similar mechanisms exist between alcohol-mediated tissue injury and that by NAFLD. We also hope that our responses are satisfactory to this Reviewer.